# Biological Methanation in an Anaerobic Biofilm Reactor—Trace Element and Mineral Requirements for Stable Operation

Joseph Tauber *, Daniel Möstl, Julia Vierheilig, Ernis Saracevic, Karl Svardal and Jörg Krampe

Institute for Water Quality and Resource Management, TU Wien, Karlsplatz 13/2261, 1040 Vienna, Austria
* Correspondence: jtauber@iwag.tuwien.ac.at

**Abstract:** Biological methanation of carbon dioxide using hydrogen makes it possible to improve the methane and energy content of biogas produced from sewage sludge and organic residuals and to reach the requirements for injection into the natural gas network. Biofilm reactors, so-called trickling bed reactors, offer a relatively simple, energy-efficient, and reliable technique for upgrading biogas via ex-situ methanation. A mesophilic lab-scale biofilm reactor was operated continuously for nine months to upgrade biogas from anaerobic sewage sludge digestion to a methane content >98%. To supply essential trace elements to the biomass, a stock solution was fed to the trickling liquid. Besides standard parameters and gas quality, concentrations of Na, K, Ca, Mg, Ni, and Fe were measured in the liquid and the biofilm using ICP-OES (inductively coupled plasma optical emission spectrometry) to examine the biofilms load-dependent uptake rate and to calculate quantities required for a stable operation. Additionally, microbial community dynamics were monitored by amplicon sequencing (16S rRNA gene). It was found that all investigated (trace) elements are taken up by the biomass. Some are absorbed depending on the load, others independently of it. For example, a biomass-specific uptake of 0.13 mg·g$^{-1}$·d$^{-1}$ for Ni and up to 50 mg·g$^{-1}$·d$^{-1}$ for Mg were measured.

**Keywords:** anaerobic digestion; biofilm reactor; biogas upgrade; ICP-OES; methanation; trace elements; trickling bed reactor

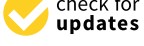



## 1. Introduction

To achieve climate and energy goals set in the Paris Agreement [1] and the EU Green Deal, renewable energy sources must be expanded dramatically within the next years. In addition, national climate and energy strategies require a significant increase in renewable energy sources like wind energy, photovoltaic (PV), biomass, and biogas. Biogas from sewage sludge and organic residuals is considered an important energy source [2,3]. In some countries, biogas, or so-called green methane gas, is seen as an important part of future green energy systems. Biogas is either used directly for electricity and heat production in combined heat and power plants or after drying, desulfurization, and removal of impurities such as dust and siloxane upgraded to biomethane, which can be used as a substitute for natural gas. Due to different legal frameworks, the quality requirements for grid injection vary between different countries [4]. Deublein and Steinhauser [5] (2011) stated that biomethane could be used as vehicle fuel or fed into the gas grid. Additionally, all the technologies required for preparing, compressing, and transporting biomethane are available and proven. Before the grid injection, gas pressure, density, sulfur, oxygen and water content, and the Wobbe Index must be adjusted [5]. Because methane gas is long-term storable and irreplaceable in some industrial high-temperature processes, like in the glass and metal industry, biomethane is seen as an important energy source in the green energy transition. For example, in Austria, the proportion of biogas should be increased from 2% in 2021 to a minimum of 10% of the national natural gas demand by 2030 [6], and energy from biomethane/biogas should increase from 0.8 TWh to 3.6 TWh from 2021 to 2030 [7]. In 2021, 260 agricultural biogas plants were in operation countrywide

and produced $108 \cdot 10^6$ m$^3 \cdot$y$^{-1}$ biogas, of which 15 plants fed biomethane into the gas grid. At the same time, 164 wastewater treatment plants (WWTPs) operated anaerobic digestion to produce $84.88 \cdot 10^6$ m$^3 \cdot$y$^{-1}$ biogas, of which three plants fed biomethane into the gas grid [8]. The biogas-based energy production can be increased in two different ways. The first is to increase the overall biogas production by digesting larger amounts of organic substrate, which is limited by the available biomass and digestion capacity and will increase $CO_2$-emissions origin from digestion and biogas usage, assuming the methane content of the biogas stays the same. It must be taken into account that the methane emissions generated during the digestion process are load-dependent [9] and would therefore increase. The second is to increase the methane content of the produced biogas using different upgrading technologies by removing or converting the $CO_2$ contained in the biogas. Kougias, et al. [10] stated that there are two ways to increase the calorific value of biogas: $CO_2$ removal or $CO_2$ upgrading. $CO_2$ removal is connected with a loss of energy and usable methane (methane slip) which is in the range of 0.5–5% depending on the technology applied [4,8]. Götz, et al. [11], Lecker, et al. [12], Rittmann, et al. [13], and Strübing, et al. [14] indicated that generating methane from $CO_2$ and externally produced $H_2$ by biological methanation can become a suitable conversion approach. Besides biological methanation using hydrogenotrophic archaea, there are a few other methods to reduce $CO_2$ contained in the biogas via physical, chemical, and biological processes. Muñoz, et al. [4], Angelidaki, et al. [2], and Fu, et al. [15] give a broad overview of available biogas upgrading methods, their advantages and disadvantages, technical properties, and costs. The most common for biogas upgrading on an industrial scale are water scrubbing, chemical scrubbing, pressure swing adsorption, and membrane separation systems. In addition, photosynthetic $CO_2$ removal by algae and immobilization in bicarbonate according to Mattiasson [16], presented in Equation (1), cryogenic processes, and chemical hydrogenation via the Sabatier process are available technologies. Chemoautotrophic biogas upgrading using hydrogenotrophic archaea, which use $CO_2$ as a carbon source by externally provided hydrogen, so-called biological methanation (BM) follows Equation (2). BM is a possibility to convert $CO_2$ to $CH_4$ via $H_2$. At the same time, $CO_2$ emissions can be reduced [17].

$$CO_2 + H_2O \rightleftharpoons H^+ + HCO_3^- \tag{1}$$

$$CO_2 + 4H_2 \rightleftharpoons CH_4 + 2H_2O \; \Delta G^0 = -130.7 \text{ kJ/mol} \tag{2}$$

In-situ and ex-situ BM have been the focus of research worldwide for several years [4]. Various reactor types operated at different temperature ranges were investigated by different authors to increase the conversion rate and the gas quality. For example, BM in continuously stirred tank reactors (CSTRs) was studied by Luo and Angelidaki [18] in a one-stage mesophilic CSTR and by Bassani, et al. [19] in a two-stage CSTR. Membrane reactors were studied by Ju, et al. [20], Luo and Angelidaki [21], Diaz, et al. [22], and Pratofiorito, et al. [23]. Submerged biofilters were investigated by Burkhardt and Busch [24], trickling bed biofilm reactors under mesophilic conditions by Burkhardt, et al. [25], Ashraf, et al. [26], and thermophilic conditions by Strübing, et al. [14], Tsapekos, et al. [27] and Feickert Fenske, et al. [28].

Ex-situ biogas upgrading in anaerobic biofilm reactors and so-called tricking bed reactors are considered relatively simple and energy-efficient methods for biological methanation [14,24,27,29]. The main advantage of (ex-situ) biofilm systems operating in the gas environment compared to systems operating in a liquid environment using fine bubbles or submerged membranes is the large surface of the biofilm carriers available for mass transfer. Based on the two-film theory, due to the low solubility of hydrogen (C) (approximately 1.5 mg/L, at 1 atm and 25 °C), the mass transfer (dC/dt) is mainly dependent on the volume-related phase interface (a) (Equation (3)), where $k_L$ is the mass transfer coefficient of the liquid phase and C* is the equilibrium concentration. The phase interface (a) cannot be determined for the bubble milieu and is combined with ($k_L$) to the volume-related mass

transfer coefficient $k_L a$. Voelklein, et al. [30] described the hydrogen-to-liquid transfer as the bottleneck of the BM process.

$$\frac{dC}{dt} = k_L \cdot a \cdot \left( C^* - C \right) = k_L a \cdot \left( C^* - C \right) \tag{3}$$

The focus in the literature on trickle bed reactors is mainly on thermophilic trickle bed reactors because of their higher conversion rates compared to systems operated under mesophilic conditions. For example, Strübing, et al. [14] operated a thermophilic (55 °C) trickle bed reactor for extended periods (313 days), achieving very high gas quality (>98% $CH_4$) at simultaneously high conversion rates of 15.4 $m^3/m^3_{trickle\ bed} \cdot d$ (corresponds to 1.56 h gas retention time). Some mesophilic trickling bed reactors can be found in the literature. Rachbauer, et al. [31] showed in long-term experiments that biogas upgrading in a mesophilic anaerobic trickling bed reactor is a feasible method for biogas upgrading. They reached stable methane concentrations >96% $CH_4$ at a gas retention time of 2.3 h. Lee, et al. [29] also performed tests in a trickling bed reactor under mesophilic conditions using bottled gas and polyurethane sponge material as biofilm carriers. They thereby achieved $CO_2$ conversion rates of 71% at a gas retention time of 2 h and 100% at a retention time of 3.8 h. Voelklein, et al. [30] performed in-situ and ex-situ tests and reached methane concentrations of 96% at gas retention times of 9.6 h in-situ and 6.5 h ex-situ. The required reactor size is thus reduced by approximately 50% in thermophilic operation compared to the mesophilic temperature range, but more heat and insulation are required for operation.

Because channeling, bypassing, and reactor dead zones can negatively affect the process performance and the product gas quality, research has focused on flow conditions in the trickle bed and optimization of the reactor shape and flow regime to increase the conversion rate. Feickert Fenske, et al. [28] investigated the co-current and counter-current operation of a trickle bed reactor and found that a co-current operation with gas flow and trickling liquid flow from top to down improves the gas residence time distribution in a plug-flow reactor. An effect of the trickling on the gas flow, depending on the applied intensity, was shown through simulations by Markthaler, et al. [32], and with measurements from Feickert Fenske, et al. [28], a discontinuous trickling was suggested to improve the methanation performance.

Because the substrate for the hydrogenotrophic biofilm is only gaseous $CO_2$ and $H_2$, essential nutrients (nitrogen and phosphorus) and trace elements must be supplied additionally via the trickling medium to guarantee stable conditions for biological methanation and biomass growth. Demirel and Scherer [33] give a broad overview of the literature on trace elements required for the digestion of agricultural and industrial substrates. They stated that the role of trace elements in anaerobic processes is significant, and the addition of a micro-nutrient solution containing Ni, Co, Mo, and Se increases the methane production and decreases the volatile fatty acid (VFA) concentration in an agricultural biogas digester. Scherer, et al. [34] reported, for example, on a range of elements in Table 1, contained in 10 methanogenic species, among others, five strains of *Methanosaracina bakeri.* The greatest variation was found for K and Na, which have important physiological functions [34]. For the enzymatic reduction from $CO_2$ to $CH_4$ via the Wolf cycle, Na, K, Fe, Ni, and Co are needed as trace metals [35]. During the hydrogenotrophic methanogenesis, $H_2$ is oxidized via hydrogenases (Ni-Fe enzymes), and ferredoxin (coenzyme $F_{420}$) is reduced [36]. Additionally, for the Wood-Ljungdahl reaction for homoacetogenic acetate production and the acetate split to $CO_2$ and $CH_4$, the trace elements Fe, Co, Ni, and S are needed [37].

Trace elements and nutrients mentioned in literature and investigated in this research can be divided into groups using the period system of elements. Alkali metals (Group I) Na and K, earth alkali metals (Group II) Ca and Mg, for which the oxidation state is precisely defined. Transition metals (Group IV) such as Mn, Fe, Co, Ni, Cu, Zn, and Mo, metals (Al), half-metals (B), and non-metals such as N, P, S, Se. Regularly adding enough trace elements is needed for stable process operation; otherwise, the methanogens will

be negatively affected. For the composition of the nutrient medium, various recipes were found in the literature, of which three from Angelidaki, et al. [38] for the digestion of cattle manure in a thermophilic CSTR, Ju, et al. [20] for a mesophilic membrane bioreactor, and Strübing, et al. [14] for a thermophilic trickling bed reactor are summarized in Table 2.

**Table 1.** Element concentrations in 10 methanogenic species, reported by Scherer, et al. [34].

| C [% w/w] | H [%] | N [%] | Na [%] | K [%] | S [%] | P [%] | Ca (Order I) [ppm] | Ca (Order II) [ppm] |
|---|---|---|---|---|---|---|---|---|
| 37–44 | 5.5–6.5 | 9.5–12.8 | 0.3–4.0 | 0.13–5.0 | 0.56–1.2 | 0.5–2.8 | 85–550 | 1000–4500 |

| Mg [%] | Fe [%] | Ni [ppm] | Co [ppm] | Mo [ppm] | Zn [ppm] | Cu [ppm] | Mn [ppm] |
|---|---|---|---|---|---|---|---|
| 0.09–0.53 | 0.07–0.28 | 65–180 | 10–120 | 10–70 | 50–630 | <10–160 | <5–25 |

The load-dependent daily dosing of potassium and sodium was calculated per liter reactor volume from values given by, and the comparatively high ammonium nitrogen concentration in the trickling liquid at Strübing, et al. [14] is striking.

Besides the trace metals and other trace compounds required for the archaeal enzymes and their biomass formation, other parameters support and direct the BM process. Fu, et al. [15] stated in a review about biogas upgrading by bioconversion that, in general, high temperature, high hydrogen partial pressure, and high ammonium concentrations promote hydrogenotrophic methanogens and inhibit acetoclastic methanogens. Zhao, et al. [39] used a stock solution containing 1 g/L $NH_4Cl$ (=337 mg/L $NH_4$) to cultivate the homoacetogen *Clostridium ragsdalei* P11 to upgrade biogas to 93–98% $CH_4$ in a membrane bioreactor. Meanwhile, Wang, et al. [40] tested the response of the biomass to different ammonia levels and showed a change in the microbial pathway from the acetoclastic pathway to the hydrogenotrophic pathway by increasing from 1 g/L to 7 g/L $NH_4$-N. Ashraf, et al. [26] investigated the dosing of nutrient medium for a thermophilic (57 °C) bio-trickling filter and stated that iron (Fe) and ammonium ($NH_4$) are crucial nutrients with concentrations of 1.5 mg/L and 0.3 g/L. Although some information about crucial trace elements for the operation of a biological methanation reactor is already given in the literature, there is a lack of information regarding trace elements for long-term stable operation, especially of mesophilic trickling bed reactors.

This study investigated the performance of an anaerobic biofilm reactor for upgrading biogas produced from municipal sewage sludge. To operate the reactor under realistic conditions, untreated digester gas was used directly as substrate (raw gas) instead of bottled gases.

The focus of this research was on the nutrient media, which is necessary for the stable operation of a mesophilic anaerobic trickling bed reactor for biological methanation. Therefore operational parameters, organic acid concentrations, and the biogas quality were investigated for different gas retention times from 1.85–18 h. Element concentrations for Na, K, Ca, Mg, Ni, and Fe were investigated in the nutrient solution, and the biomass was attached to the biofilm carriers during nutrient solution dosing experiments. In addition, the biomass concentration and the composition of the microbial community in the biomass on the biofilm carriers were studied as a function of position in the trickle bed reactor, and microscopic images were taken from two different biofilm carriers during eight months of continuous operation. The study shows the biomass (mass) and reactor volume-specific nutrient requirements for the stable operation of a trickle bed reactor for a given raw gas load and conversion rate and suggests the required dosing quantities.

**Table 2.** Comparison of elements and compounds for dosing in the trickling liquid for three references Angelidaki, et al. [38], Ju, et al. [20], and Strübing, et al. [14]. Chemical compound and concentration of trace elements (mass referred to as element/compound X) in the stock solution and the reactor/trickling liquid in mg/L.

| Reference Element X | Ju et al. [20] Compound | | Element | Angelidaki et al. [38] Compound | | Element | | Strübing et al. [14] Compound | | Element |
| --- | --- | --- | --- | --- | --- | --- | --- | --- | --- | --- |
| [Name] | [Formula] | [mg/L] Membrane Reactor | [mg X/L] Membrane Reactor | [Formula] | [mg/L] Stock Solution | [mg X/L] Stock Solution | [mg X/L] Reactor | [Formula] | [mg/L] Stock Solution | [mg X/L] Trickling Liquid |
| Mg | $MgCl_2 \cdot 6H_2O$ | 16.05 | 1.919 | $MgCl_2 \cdot 6H_2O$ | 10,000 | 1195.512 | 12.274 | $MgCl_2 \cdot 6H_2O$ | 300 | 35.865 |
| Ca | $CaCl_2 \cdot 2H_2O$ | 1.2 | 0.327 | $CaCl_2 \cdot 2H_2O$ | 5000 | 1363.068 | 13.994 | - | - | - |
| Zn | $ZnCl_2$ | 5.91 | 2.835 | $ZnCl_2$ | 50 | 23.988 | 0.0246 | - | - | - |
| Mo | $Na_2Mo \cdot 2H_2O$ | 1.29 | 0.695 | $(NH_4)6Mo_7O_{24} \cdot 4H_2O$ | 50 | 27.171 | 0.0279 | $(NH_4)_6Mo_7O_{24} \cdot 2H_2O$ | 1.5 | 0.84 |
| Se | - | - | - | $Na_2SeO_3 \cdot 5H_2O$ | 100 | 36.723 | 0.0377 | $Na_2SeO_3 \cdot 5H_2O$ | 0.1 | 0.0367 |
| Ni | - | - | - | $NiCl_2 \cdot 6H_2O$ | 92 | 22.718 | 0.0233 | $NiCl_2 \cdot 6H_2O$ | 9 | 2.222 |
| Mn | $MnCl_2 \cdot 4H_2O$ | 13.19 | 3.662 | $MnCl_2 \cdot 4H_2O$ | 50 | 13.88 | 0.0143 | - | - | - |
| Cu | $CuCl_2 \cdot 2H_2O$ | 2.61 | 0.973 | $CuCl_2 \cdot 2H_2O$ | 38 | 14.164 | 0.0145 | - | - | - |
| Co | $CoCl_2 \cdot 6H_2O$ | 0.3 | 0.0801 | $CoCl_2 \cdot 6H_2O$ | 50 | 12.385 | 0.0127 | $CoCl_2 \cdot 6H_2O$ | 1.5 | 0.372 |
| K | KCl | 1 | 0.524 | $K_2HPO_4 \cdot 3H_2O$ | 200,000 | 34,255.871 | 70.330 | $K_2HPO_4$ | load depending (60–200 mg/$L_{reactor} \cdot d$) | |
| Fe | $FeCl_2 \cdot 2H_2O$ | 5.23 | 1.794 | $FeCl_2 \cdot 4H_2O$ | | 0.686 | 0.000705 | $FeCl_2 \cdot 4H_2O$ | 750 | 210.678 |
| EDTA | EDTA | 9.75 | 9.75 | EDTA | 500 | 500 | | EDTA | 750 | 750 |
| Na | NaCl | 200 | 78.678 | | 10,000 | 3933.949 | 40.389 | $Na_2CO_3$ | 9000 | 1952.168 |
| P | $(NH_4)_2HPO_4$ | 200 | 46.909 | $K_2HPO_4 \cdot 3H_2O$ | 200,000 | 27,140.702 | 55.722 | - | - | - |
| B | - | - | - | $H_3BO_3$ | 50 | 8.743 | 0.00898 | - | - | - |
| Al | - | - | - | $AlCl_3$ | 50 | 10.118 | 0.0104 | - | - | - |
| S | - | - | - | Cystein hydrochloride $C_3H_7NO_2S$ | 500 | 98.494 | 1.001 | $Na_2S \cdot 9H_2O$ | load depending (18–300 mg/$L_{reactor} \cdot d$) | |
| N | $(NH_4Cl)_2HPO_4$ | 200 | 21.213 | $NH_4Cl$ | 100,000 | 26,185.227 | 268.842 | $NH_4Cl$ | 7300 | 1911.522 |

## 2. Materials and Methods

### 2.1. Lab Scale Experiments

#### 2.1.1. Biofilm Reactor Setup

A lab-scale biofilm reactor with a working volume of 8 L was operated continuously for 8 months (225 days) to upgrade biogas using biological methanation at mesophilic conditions (37 °C). The reactor was designed as a three-phase system, where hydrogenotrophic biomass forms a biofilm-on-biofilm carriers and produces methane according to Equation (2) of carbon dioxide and hydrogen that is fed into the reactor. The biomass is supplied with water, nutrients, and trace elements by trickling liquid. The raw gas was produced from a lab-scale anaerobic digester with a working volume of 14 L. As substrate for the digester, raw sludge (a mixture of primary- and waste-activated sludge) from a municipal wastewater treatment plant in Austria was digested. The untreated digester gas was directly supplied to the reactor in a bottom-up flow direction. The trickling liquid with the nutrient and trace element solution was supplied in counter-current flow from top to bottom. The biofilm reactor for the gas upgrade with its main components is presented in Figure 1. The trickle bed was filled with two different biofilm carriers (Figure 1b), with a volume of 4 L each. The Linpor® polyurethane cubes with an edge length of 12–15 mm and a specific surface of 2000 m²/m³ specified by the manufacturer (Strabag Water Technologies, Vienna, Austria) [41] were used in the lower area of the trickle bed. The porosity of the cubes is reported to be 97% at 40 pores per 25.4 mm (1 inch). In the upper area of the trickle bed, the Xingfeng PE-10 polyethylene carriers (25 mm diameter, 4 mm thick) with a specific surface of 1200 m²/m³ and a bulk density of 140,000 pcs/m³ [42] was used. The biofilm carriers were inoculated with 530 g of digester sludge from a lab-scale anaerobic digester with a hydraulic retention time of 30 days, operated at mesophilic conditions (38 °C), and fed with raw sludge (a mixture of primary- and waste-activated sludge) from a municipal WWTP in lower Austria. Digester sludge from this WWTP also served as reverence for the DNA analysis of the inoculum and the biofilm reactors biomass.

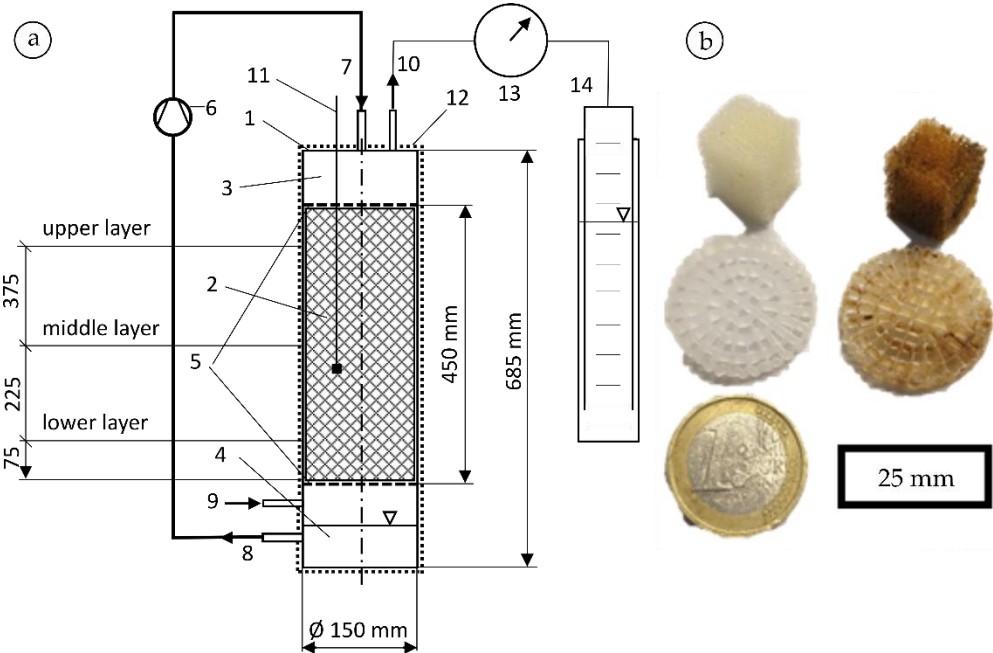

**Figure 1.** (**a**) Scheme of the biofilm reactor, including three (upper, middle, lower) sampling layers: (1) biofilm reactor, (2) trickle bed 8 L bulk volume, (3) gas compartment 1 L volume, (4) reservoirs of liquid 1 L volume, (5) wire mesh, (6) liquid circulation pump, (7) liquid inlet, (8) liquid outlet, (9) biogas inlet (raw gas from anaerobic digester), (10) upgraded gas outlet, (11) temperature sensor, (12) electric heating jacket and insolation, (13) drum gas counter, (14) swimming gas storage. (**b**) Linpor® PU-foam cube and PE-10 biofilm carriers, left new, right after biofilm formation (day 105).

The inoculum was sieved with a mesh width of 1 mm to remove particles to prevent mechanical damage and clogging of the pipes. For the inoculation, the inoculation sludge was pumped over the biofilm carriers in a circle for 5 days using the liquid circulation pump, after which the excess sludge was removed from the liquid reservoir. At the beginning of the inoculation phase, the biofilm reactor was flushed with five times the reactor volume of methane to guarantee anaerobic conditions. After each reactor opening for sampling, the reactor also was flushed with methane.

The $CH_4$, $CO_2$, $O_2$, $H_2$, and $H_2S$ concentration in the gas in- and outlet was measured daily, and during the gas flushing of the reactor using a hand probe (Gas Data GFM 430, Shawcity Limited, Coventry, England), the measurement accuracy is specified as 3% by the manufacturer at 100% $CO_2$ and $CH_4$. The gas volume was measured using a drum gas counter (TG0.5/5 PVC, Dr.-Ing. Ritter Apparatebau GmbH, Bochum, Germany). The reactor temperature was measured and controlled using a platin temperature sensor and temperature controller (STC-1000 Pro, Diymore, Shenzhen, China) connected to an electrical heat jacket (HK-5.0, 75 W, Arnold Rak Wärmetechnik, Frankfurt, Germany).

The reactor was supplied with biogas from an experimental plant; the raw biogas composition and the product gas composition was measured daily. The hydrogen required for methanation was produced in a PEM electrolysis (hydrogen generator 20 H, Parker Hannifin GmbH, Bielefeld, Germany). The dosage was controlled via a needle valve and a variable area flow meter (rotameter) calibrated for hydrogen (P16A1BA0, 65 mm, 0–35 mL/min, Aalborg Instruments & Controls, Inc., New York, NY, USA). The hydrogen was introduced in a stoichiometric ratio to the carbon dioxide of 4:1, as shown in Equation (2). The energy requirement for methane production via biological methanation, including efficiencies, can be given as 22.5 kWh/$m^3$ methane, according to calculations by Tauber, Ramsbacher, Svardal and Krampe [8], not taking into account the oxygen produced via electrolysis. An efficiency of 49–79% for the whole power-to-gas process via biological methanation is indicated by Sterner and Stadler [43], depending on the pressure level, storage type, and technology used. For the trickling liquid circulation, a peristaltic pump (323S, Watson Marlow, Guntramsdorf, Austria) with a 6.4 mm diameter tube was used.

### 2.1.2. Trickling Liquid and Trace Element Dosing

The trickling liquid was circulated continuously with a volume flow of 30 mL/min. A trace element stock solution based on the compositions from literature (Table 2) was prepared following the composition in Table 3 and added to the trickling liquid. Because raw biogas from a digester was used as the substrate, no extra sulfur was added. On average, the raw gas contained 205 ppm $H_2S$, corresponding to a daily sulfur input into the reactor of 600–2000 mg/d, depending on the raw biogas volume flow.

### 2.1.3. Biomass Sampling

The biomass was sampled at three different positions in the reactor, as shown in Figure 1. The lower layer was located 75 mm above the ground of the trickle bed, the middle layer was in the middle of the trickle bed, and the upper layer was located 375 mm above the ground of the trickle bed. The biomass was also sampled over the entire experiment duration to calculate the biomass growth and decay rate.

### 2.2. Analytical Methods
### 2.2.1. Chemical Standard Analysis

Chemical oxygen demand (COD), total Kjeldahl nitrogen (TKN), total nitrogen (TN), ammonium-nitrogen ($NH_4$-N), and orthophosphate-phosphorus ($PO_4$-P) concentrations were measured according to relevant standard procedures (Table 4). Organic acid concentrations of formic-, acetic-, propionic-, butyric-, lactic-, and valeric-acid in the trickling liquid have been measured by High-Performance Liquid Chromatography (HPLC) after filtration using a two-stage syringe filter with 1 μm and 0.45 μm pore wide (Chromafil

GF/RC-45/25 Machery-Nagel, Düren, Germany) and stored at 4 °C until measurement. Each trickling liquid sample was measured three times.

**Table 3.** Element and compound concentrations in the stock solution used for dosing and the resulting concentration in the trickling liquid.

| Element X [Name] | Compound [Formula] | [mg/L] | Element [mg X/L] |
|---|---|---|---|
| | | Stock Solution | Trickling Liquid |
| Mg | $MgCl_2 \cdot 6H_2O$ | 300 | 35.865 |
| Ca | $CaCl_2 \cdot 2H_2O$ | 51.308 | 13.994 |
| Zn | $ZnCl_2$ | 0.0514 | 0.0246 |
| Mo | $(NH_4)_6Mo_7O_{24} \cdot 2H_2O$ | 1.5 | 0.84 |
| Ni | $NiCl_2 \cdot 6H_2O$ | 9 | 2.222 |
| Mn | $MnCl_2 \cdot 4H_2O$ | 0.0514 | 0.0143 |
| Cu | $CuCl_2 \cdot 2H_2O$ | 0.0390 | 0.0145 |
| Co | $CoCl_2 \cdot 6H_2O$ | 9 | 2.228 |
| K | $K_2HPO_4 \cdot 3H_2O$ | 410.614 | 70.338 |
| Fe | $FeCl_2 \cdot 4H_2O$ | 750 | 211 |
| EDTA | EDTA | 750 | |
| B | $H_3BO_3$ | 0.0514 | 0.00898 |
| Al | $AlCl_3$ | 0.0513 | 0.0104 |
| Na | NaCl | 102.679 | 40.393 |
| N | $NH_4Cl$ | 7300 | 1912 |
| P | contained in $K_2HPO_4 \cdot 3H_2O$ | | 27.955 |
| S | supply via $H_2S$ in the biogas | | 600–2000 mg/d |
| Se | not contained | - | - |

**Table 4.** Applied methods for chemical standard analysis.

| Parameter | Method | Comment |
|---|---|---|
| COD | DIN 38409 | DEV H 43-1 Short-term method |
| TKN, TN | DIN 19684 part 4 | DEV H11 Photometric variant |
| $NH_4$-N | DIN 38406 | DEV D5-1 Photometric |
| $PO_4$-P | DIN 38405 | DEV D11-3 Photometric |
| organic acids | DIN EN 17294 | Aminex HPX-87 H column, 5 mN $H_2SO_4$ mobile phase, UV-detector at 625 nm |
| DM, oDM | DIN 38409 part 1 | DEV H1 Annealing temperature 550 °C |
| SS | DIN 38409 part 2 | |
| pH-value | DIN 38404 part 15 | DEV C5 WTW SenTix20 combination electrode |

2.2.2. Dry Matter- and Organic Dry Matter Concentration of the Biofilm Carriers

To measure the biomass concentration attached to the biofilm carriers, at each sampling time, ten Linpor® PU-foam cubes were squeezed out and washed 5–6 times with deionized water according to the method of Yuan [44]. Then the suspension was filtered using an ashless filter with 0.45 μm pore size (cellulose nitrate filter, Satorius Stedim Biotech GmbH, Göttingen, Germany) to preserve the biomass. To determine the solid dry mass (DM) and the organic dry mass (oDM), the filter was dried and ashed according to DIN 38409 part 1. With the number of biomass carriers and the bulk volume, the volume-specific biomass concentration was calculated.

2.2.3. Biomass and Trickling Liquid Sample Revealing and Nutrient Analysis (ICP-OES)

The biofilm carriers and the trickling liquid were sampled regularly to measure concentrations of Na, K, Ca, Mg, Ni, and Fe using ICP-OES (inductively coupled plasma optical emission spectrometry). Therefore, the biofilm carriers and the trickling liquid were treated as described above. To the dried filter, including the extracted biomass, each of 2 mL $H_2SO_4$

(concentrated), 7 mL of $HNO_3$ (concentrated), and 3 mL $H_2O_2$ (30%) were added and then made up to 50 mL volume with deionized water. The suspension was then disintegrated at 210 °C for 15 min by microwave (START microchemist 1500, MLS-MWS Laboratory solutions, Leutkirch, Germany). Subsequently, the samples were measured with an Optima 8300 ICP-OES Optical System and SCD Detector (Perkin Elmer Inc, Waltham, MA, USA).

The elements (Na, K, Ca, Mg, Ni, and Fe) were determined using axial view and triple determination, followed by arithmetic averaging. For calibration, a customized single-element (Merck, Roth, Germany) standard was applied. Biomass-specific trace element concentrations were calculated using the measured trace element concentrations in the biomass and the DM- and oDM-concentration of the biomass.

From the rate of change of concentration in the trickling liquid, the specific rate of decrease in mg/L·d was calculated for each element. From this, the trace elements required per volume (liter) and month were calculated.

The suspended solids (SS) concentration in the trickling liquid was measured regularly according to DIN 38409-2.

### 2.2.4. Microscopic Imaging and Amplicon Sequencing (16S rRNA Gene)

The biofilm carriers were regularly sampled, and the biomass was examined using a microscope (Leica DMRE). Therefore, the biofilm carriers were cut into thin slices, and photos were taken in transmitted light and phase contrast mode.

For Illumina MiSeq-based highly multiplexed 16S rRNA gene amplicon sequencing, biofilm carriers were cut into small pieces and centrifuged for 10 min at $4500\times g$. The supernatant was removed, and the samples were stored at −20 °C until total nucleic acids (TNA) were extracted with the DNeasy PowerSoil Pro Kit from QIAGEN. The TNA extraction, 16S rRNA gene amplicon sequencing, and bioinformatic analysis were conducted by the Joint Microbiome Facility of the Medical University of Vienna and the University of Vienna (JMF) under project ID JMF-2212-06. For amplicon sequencing, the V4 region of the 16S rRNA gene was amplified using the primers 515F (5′-GTG CCA GCM GCC GCG GTA A-3′) [45] and 806R (5′-GGA CTA CNV GGG TWT CTA AT-3′) [46]. Subsequently, barcoding using a dual barcoding approach, amplicon normalization, sequencing library preparation, sequencing on an Illumina MiSeq, sequence processing, and sequence analysis were performed as described in Pjevac, et al. [47]. Amplicon sequence variants (ASVs) were inferred using the DADA2 R package [48,49], and ASVs subsequently classified using the classifier implemented in DADA2 v.1.26.0 and the SILVA database SSU Ref NR 99 release 138.1 using default parameters [50,51]. Selected ASVs were subjected to BLAST searches against NCBI's nt (nucleotide collection) sequence database [52–54]. The 16S rRNA gene amplicon sequencing data have been deposited at the NCBI's Sequence Read Archive under the BioProject accession number PRJNA947948.

## 3. Results and Discussion

### 3.1. Raw Gas Composition

The raw biogas composition at the input to the trickling bed reactor is presented in Table 5. Hydrogen was additionally dosed, with the amount manually controlled by a needle valve as described above.

**Table 5.** Raw biogas composition at the input to the trickling bed reactor, n = 139 samples.

| $CH_4$ Mean ± SD [%] | | $CO_2$ Mean ± SD [%] | | $O_2$ Mean ± SD [%] | | $H_2S$ Mean, min/max [ppm] | | $H_2$ [%] | $H_2O$ [%] | Temperature Mean ± SD [°C] | |
|---|---|---|---|---|---|---|---|---|---|---|---|
| 65.4 | ±2.83 | 28.7 | ±4.55 | 0.4 | ±0.21 | 205 | 0–1958 | 0 | 6.52 | 37.9 | ±0.16 |

### 3.2. Product Gas-Flow and -Composition, Reactor Temperature, Ammonium Concentration, Nitrogen Demand, and Acetic Acid Concentration in the Biofilm Reactor

In Figure 2a, the product gas flow of the biofilm reactor and the methane and carbon dioxide concentrations measured in the product gas is displayed. The gas retention time

(GRT) depended on the bed volume and the gas volume flow and varied between 18 h and 1.85 h. The bed volume was gradually reduced from 8 L to 2 L to increase the gas load and then increased again to 4 L at phase III (start-up 2) by adding new biofilm carriers on day 174 to test the start-up behavior (Figure 2b).

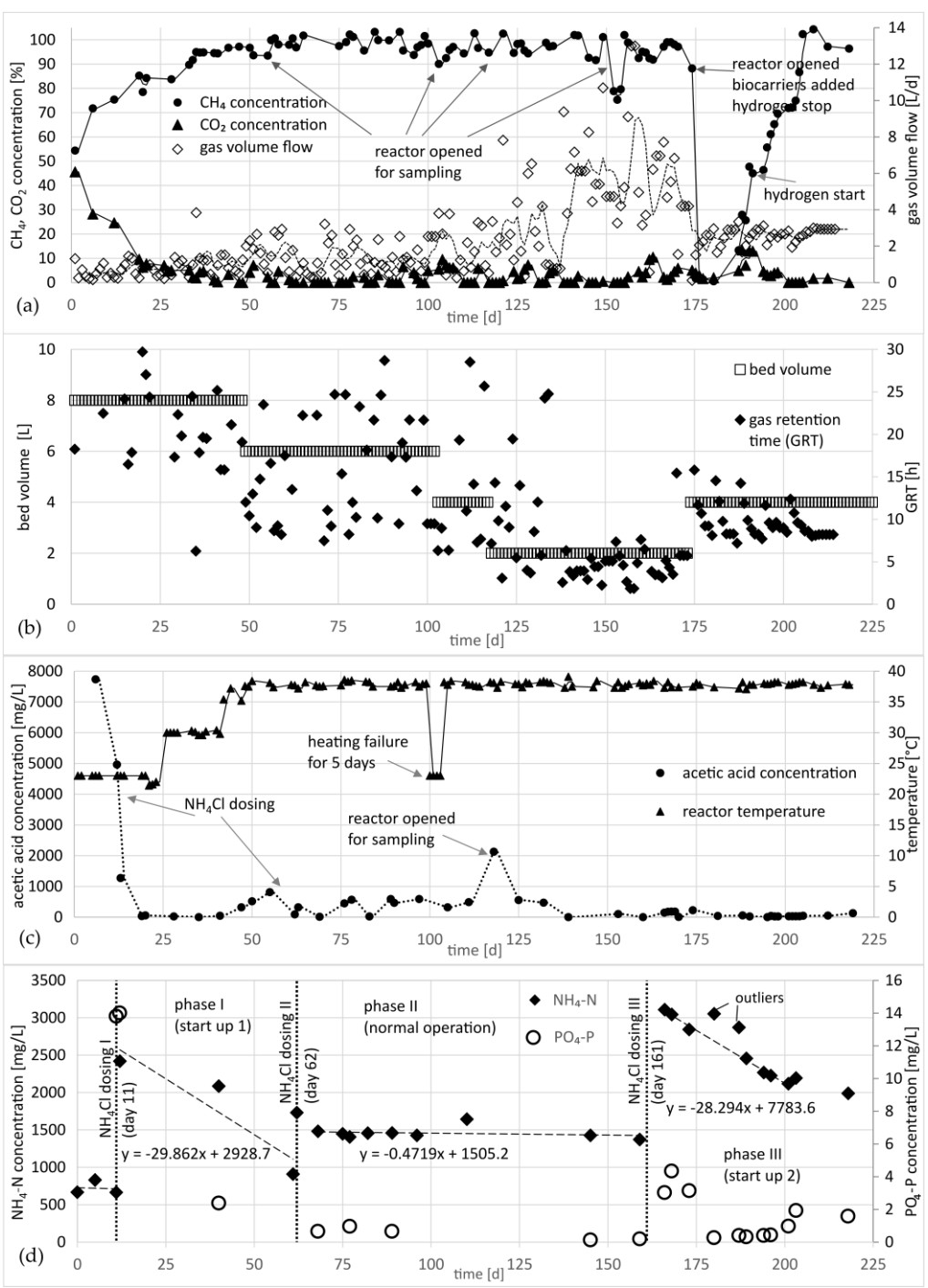

**Figure 2.** (**a**) Product gas volume flow and methane and carbon dioxide concentration. (**b**) Bed volume and gas retention time of the biofilm reactor. (**c**) Reactor temperature and acetic acid concentration in the trickling liquid. (**d**) $NH_4$-N and $PO_4$-P concentration in the trickling liquid, $NH_4Cl$ dosing I, II, and III are marked with dotted lines. Mean concentration rates for different phases are marked with dashed lines.

The gas flow was increased from approximately 2 L/d at the beginning to 13 L/d at the end of the experiment (Figure 2a). During the start-up phase, the methane concentration in the product gas increased from 55% on day 1 to 85% on day 19, while the acetic acid concentration decreased from 7735 mg/L to 36 mg/L (Figure 2c).

On day 12, 11.3 g $NH_4Cl$ (equal to 2.97 g $NH_4$-N) was dosed into the trickling liquid, and the $NH_4$-N concentration increased from 664 mg/L to 2421 mg/L. Within one day, the acetic acid concentration dropped from 4964 mg/L to 1276 mg/L. As presented in Figure 2c, on day 26, the heating was turned on, and the reactor temperature was set to 30 °C. On day 44, the temperature was set to 38 °C. After a continuous increase of the acetic acid concentration from day 35 (4 mg/L) to 813 mg/L on day 55, a second time $NH_4Cl$ (12.3 g equal to 3.22 g $NH_4$-N) was dosed into the trickling liquid to increase the $NH_4$-N concentration from 908 mg/L (day 62) to 1730 mg/L (day 63). After increasing the bed volume on day 174 from 2 L to 4 L by adding new Linpor® biofilm carriers, the hydrogen supply was paused for 17 days and started again on day 191. After this standstill at normal operation temperature (warm shutdown), the methane concentration reached >95% again after 14 days on day 205. The product gas composition and operational parameters, such as pH value and the reactor temperature during the normal operation phase, are displayed in Table 6.

**Table 6.** Product gas composition and operational parameters during the normal operation phase of the biofilm reactor (days 62–160).

| $CH_4$ Mean ± SD [%] | $CO_2$ Mean ± SD [%] | $O_2$ Mean ± SD [%] | $H_2S$ Mean ± SD [ppm] | $H_2$ Mean ± SD [ppm] | pH-Value Mean ± SD [-] | Temperature Mean ± SD [°C] |
|---|---|---|---|---|---|---|
| 96.6 ± 5.91 | 1.6 ± 2.42 | 0 ± 0.00 | 857 ± 861 | 19,169 ± 13,689 | 7.81 ± 0.45 | 36.9 ± 3.77 |

The reactor temperature dropped from 37 °C to about 22 °C for 5 days due to a fault in the heater (Figure 2b), but this did not cause any noticeable change in the gas quality. The pH value in the tricking liquid was pH 7.81, within the expected range.

The ammonium-nitrogen and orthophosphate-phosphor concentrations in the trickling liquid, as well as the $NH_4Cl$ dosings, are displayed in Figure 2d). Concentration rates for $NH_4$-N of 29.8 mg/L·d and 28.3 mg/L·d were calculated for the two start-up phases (I and III), and 0.5 mg/L·d was calculated for the phase of constant operation, phase II (days 62–160). The difference shows that nitrogen is needed for biomass build-up during the start-up phases, whereas relatively little nitrogen is needed for the metabolism during normal operation.

*3.3. Organic Acid Concentrations in the Trickling Liquid*

The concentrations for acetic-, formic-, propionic-, lactic-, butyric-, and valeric acid are displayed in Figure 3. During the start-up phase till day 20, the acetic acid concentration decreased from 8000 mg/L to below 100 mg/L after dosing trace element solution and $NH_4$-N on day 12. Banks, et al. [55] reported that organic acid concentrations below 500 mg/L indicate stable operation of anaerobic digesters, and high concentrations in predominated hydrogenotrophic populations can be prevented by dosing trace elements.

The composition of organic acids changed from mainly acetic acid at the start-up phase to a mixture of acetic, butyric-, and valeric acid when the hydrogenotrophic methanogens were dominant, and the methane concentration increased to above 90% $CH_4$ Figure 3b. During the constant operation phase, the organic acid concentration was in total in the range of 300–900 mg/L. Notable was the increase of the valeric acid concentration to approximately 100 mg/L from day 75 on and the decrease to a concentration of approximately 50 mg/L valeric acid after the third ammonium dosing. It is assumed that the metabolic pathway is partially switched from hydrogenotrophic methanation to the formation of valeric acid at low ammonium concentrations. Very high organic acid concentrations, which can be explained by the opening of the reactor and a reduction of the bed volume from 4 L to 2 L,

were detected on day 117. After the third ammonium dosing on day 161, the organic acid concentration was relatively stable between 60 mg/L and 360 mg/L.

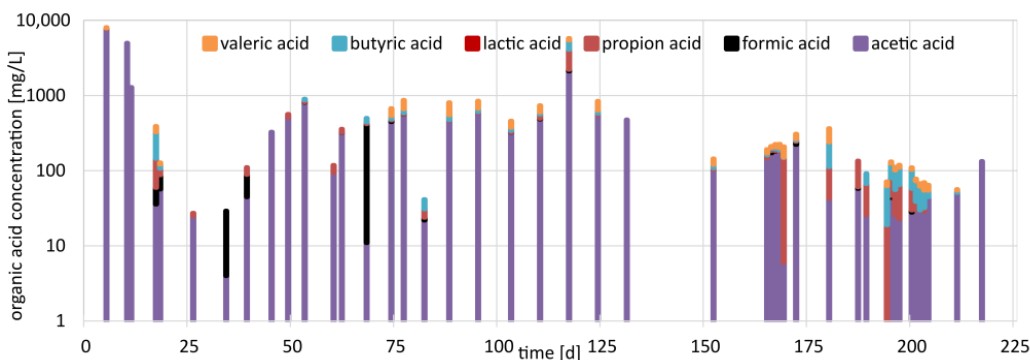

**Figure 3.** Organic acid concentrations in the trickling liquid, in logarithmic representation.

### 3.4. Trace Element Concentration in the Trickling Liquid and Biomass

The trace element concentrations in the trickling liquid and the biomass were measured in short time intervals between days 182 and 218. The concentrations measured in the biomass are displayed in Figure 4. After dosing trace element stock solution on day 196, the gas loading of the trickle bed reactor was kept nearly constant for one week till day 204. Then the gas supply was reduced to very low levels by stopping the raw sludge supply of the anaerobic digester from which the raw gas is produced. Until day 220, the gas load of the trickle bed reactor was relatively low at 0.5 L/L·d. The TSS concentration in the trickling liquid was in the range of 14–96 mg/L and relatively constant during the whole experiment.

### 3.5. Nutrient Demand of the Biomass

The biomass from the biofilm carriers was sampled nine times, and the concentrations for Fe, Mg, Ca, Na, K, and Ni were measured using ICP-OES. On day 196, an amount of 100 mL nutrient media according to the recipe (stock solution in Table 3) was supplied to the trickling liquid. In the following eight days, the reactor was operated at constant conditions with a raw gas flow of 2.5 L/L·d and a gas retention time of GRT = 9.1 h, respectively. The BMR was 1.35 L/L·h on average. In this timeframe, the output gas quality increased from 65% to 100% $CH_4$. On day 204, the hydrogen supply was stopped. The trace element concentration in the unfiltered trickling media for Ni, Fe, P (as $PO_4$-P), K, Na, Mg, and Ca for both phases (high load and pause) are displayed in Figure 4.

For nickel, the starting concentration after the dosing was 8 mg/L. The uptake rate during the high-load phase was 0.13 mg/L · d, the same as during the pause.

Different uptake rates for potassium, sodium, magnesium, and calcium can be distinguished for the high-load phase and the break.

The trace element concentration after the dosing, the specific daily demand, and a theoretical range are displayed in Table 7. Also, the trace elements released by biomass decay and the monthly dosing quantity required to keep the concentration constant are shown in Table 7.

### 3.6. Trace Element Concentration in the Biomass according to Sampling Position and Time

The trace element concentrations in the biomass, sampled at three different positions on day 257, are displayed in Figure 5. All concentrations decrease from the lower to the upper sampling positions, except for potassium, which the decreasing gas load and the associated greater biomass growth from the bottom to the top of the trickling bed can well explain.

**Table 7.** Element concentration after dosing, specific daily demand, range after dosing, the amount released by biomass decay, and monthly dosing amount for stable concentrations for Fe, Ni, K, Na, Ca, and Mg.

| Trace Element | Start Concentration | Demand | Range | Decay | Monthly Dosing |
|---|---|---|---|---|---|
| Name | [mg/L] | [mg/(L·d)] | [d] | [mg/(L·d)] | [mg/(L·mo)] |
| Fe | 28 | - | - | - | - |
| Ni | 8 | −0.13 | 61 | −0.13 | 4 |
| K | 230 | −6.95 | 33 | 0.20 | 210 |
| Na | 180 | −7.67 | 23 | 0.58 | 230 |
| Ca | 330 | −18.02 | 18 | −0.51 | 540 |
| Mg | 1230 | −49.15 | 25 | −1.18 | 1475 |

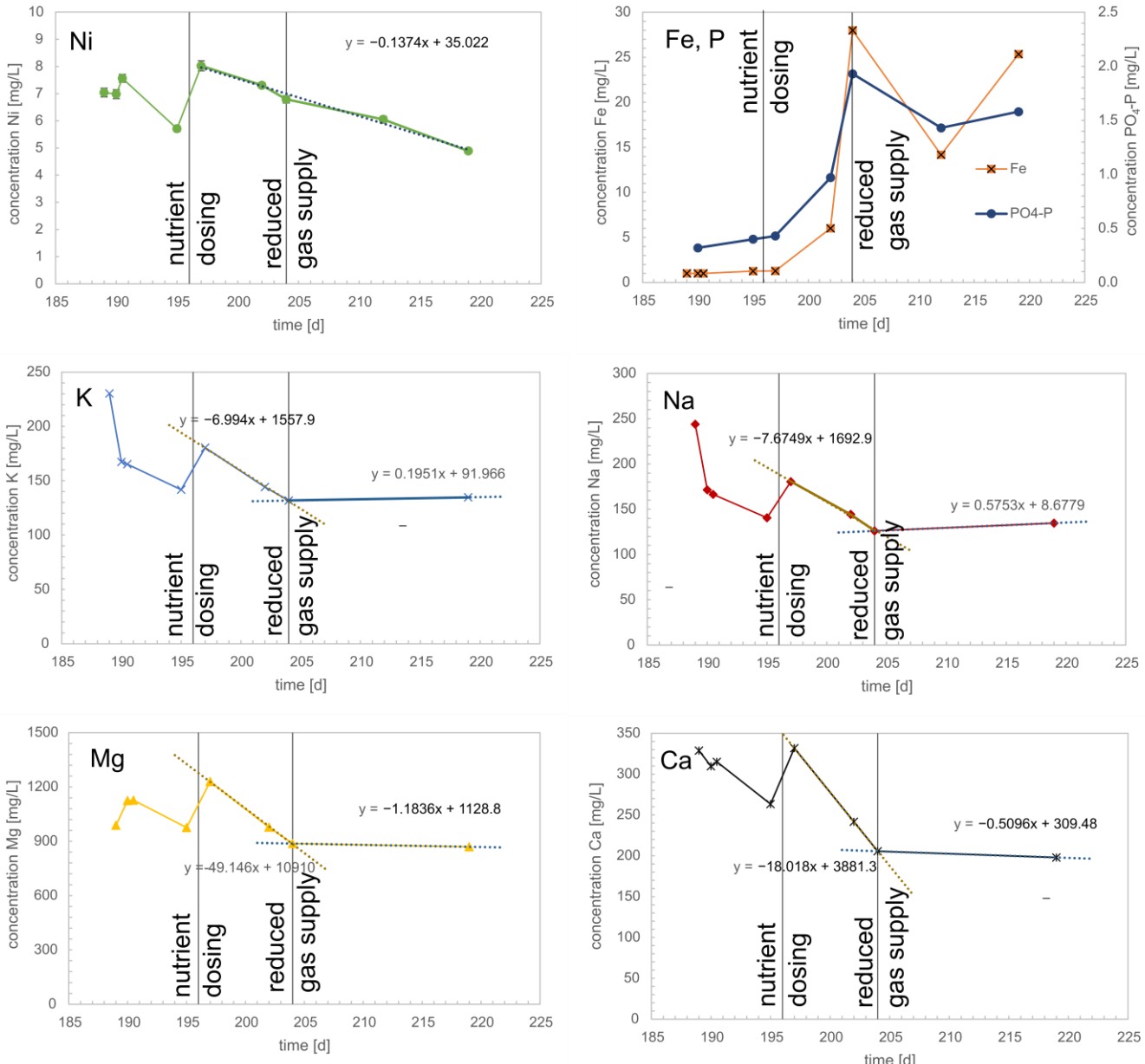

**Figure 4.** Trace element and mineral concentration in the trickling media and demand for Ni, Fe, P (as PO$_4$-P), K, Na, Mg, and Ca after dosing on day 196, the trend lines for days 197–204 and days 204–219 are drawn as dotted lines.

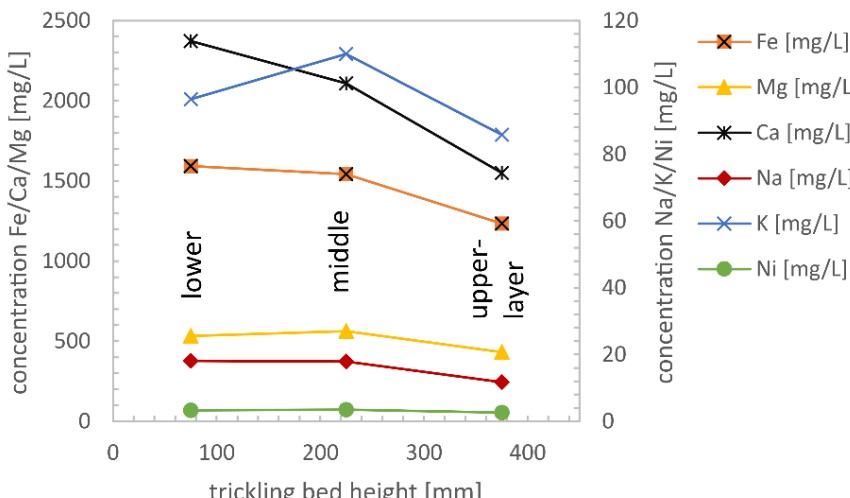

**Figure 5.** Element concentration in the biomass according to the sampling position (day 257).

*3.7. Biomass Concentration in the Trickling Bed*

The dry matter (DM) and organic dry matter (oDM) concentration in the trickling bed is displayed in Figure 6. DM and oDM concentrations decrease linearly from the lower layer to the upper layer of the reactor (Figure 6a), which can be explained by the higher gas conversion rate at the raw gas inlet at the reactor bottom and the lower conversion rate at the product gas outlet. In Figure 6b, the DM and oDM concentrations increase can be seen. For days 102–218, a linear regression was calculated. The oDM concentration increased by 106 mg/L·d, while the DM concentration increased by 223 mg/L·d. After stopping the gas supply, the oDM concentration decreased by 35 mg/L·d, and the DM concentration decreased by 51 mg/L·d due to biomass decay.

The suspended solids (SS) concentration in the trickling liquid was between 14 and 96 mg/L.

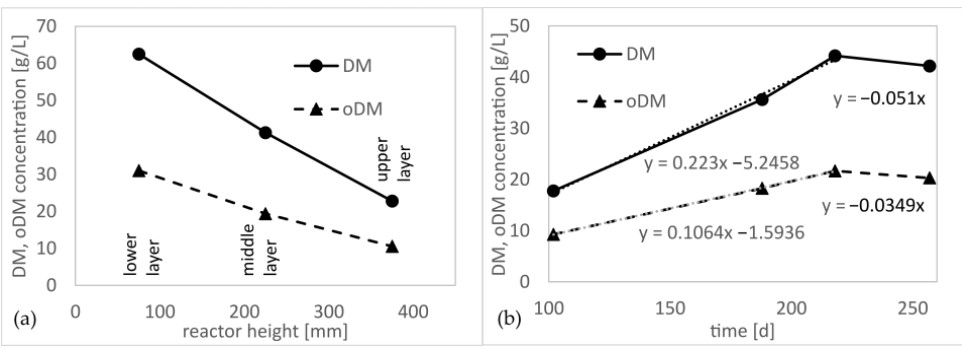

**Figure 6.** Dry matter (DM) and organic dry matter (oDM) concentration in the reactor, (**a**) depending on the position for day 257 and (**b**) depending on time, rates for days 102–218 are marked.

*3.8. Microscopy Imaging*

Biofilm carriers were sampled, and microscopic images were taken regularly during the whole reactor operation. Figure 7 presents images of the biomass on the PU-foam biofilm carriers at different times and locations in the reactor. On day 102, a thin biofilm was seen on all carriers examined (Figure 7a) Visually, no difference was detectable between the individual sampling sites (Figure 7b) upper layer, (Figure 7c) middle layer, and (Figure 7d) lower layer on day 257 and the sample from day 102.

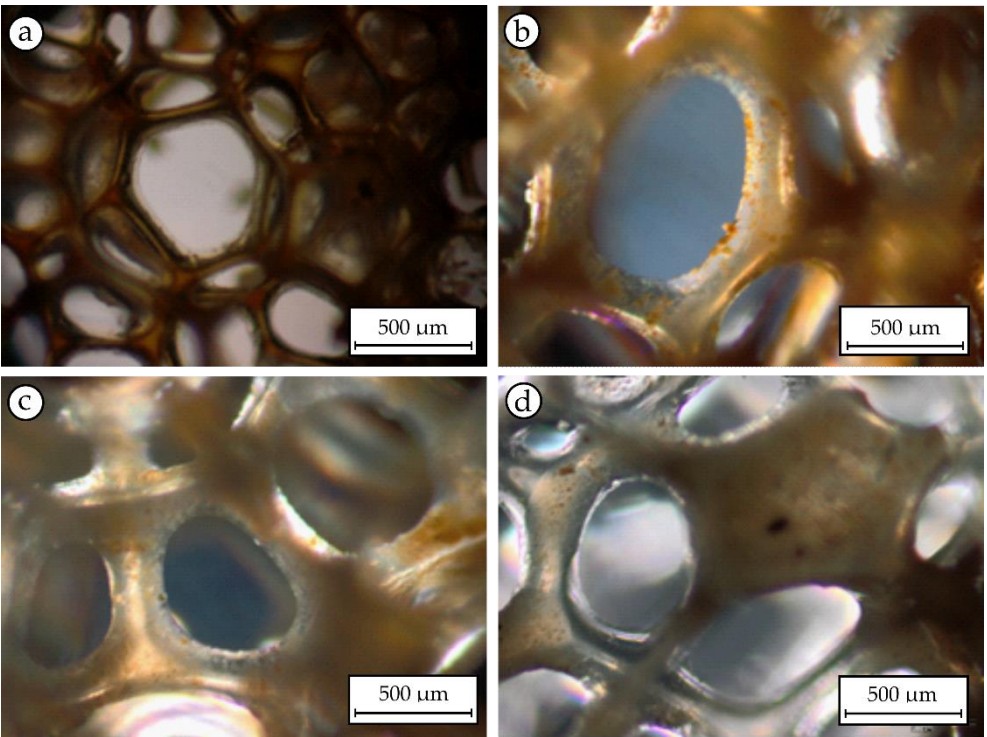

**Figure 7.** Microscopic images of the biofilm on the Linpor® biofilm carriers, according to the sampling position and sampling time (**a**) upper layer (day 102), (**b**) upper layer (day 257), (**c**) middle layer (day 257), (**d**) lower layer (day 257).

Microscopic images of the biomass on the PE10 biofilm carriers sampled on day 102 (Figure 8a) and day 257 (Figure 8b) are presented. As with the PU-foam cubes, no visual difference was noticeable between the two time points. At both times, a biofilm about 5–10 μm thick covered the entire biofilm-bearing surface.

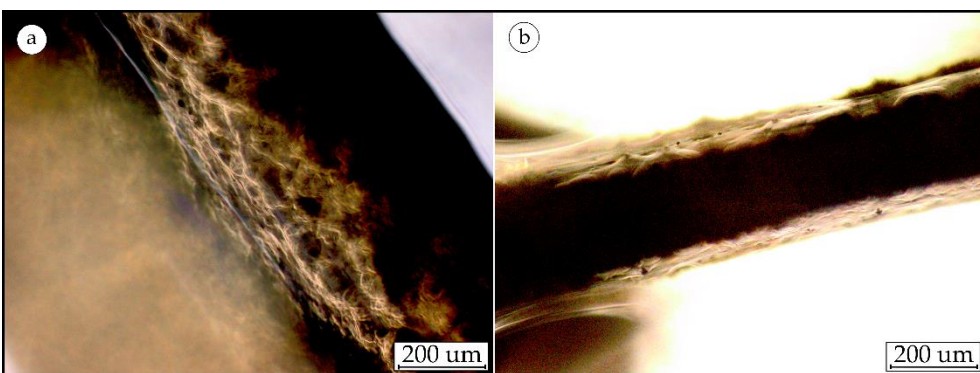

**Figure 8.** Microscopic images of the biofilm on the PE10 biofilm carriers, sampling position upper layer (**a**) on day 102, (**b**) on day 257.

### 3.9. Microbial Community Dynamics of the Biomass

An overview of the microbial community dynamics on the phylum level, analyzed by 16S rRNA gene amplicon sequencing (Illumina MiSeq, San Diego, CA, USA), is presented in Figure 9. The sludge which served as inoculum for the biofilm reactor and a digester sludge that served as a reference showed a diverse mixture of 24 and 27 different phyla, respectively. The most abundant phyla (>10% relative abundance in at least one sample) are shown in Figure 9. The relative abundance of *Proteobacteria* and *Euryarchaeota* in the biofilm increased from 3.8% to up to 13.4% (upper layer) and from 1.1% to up to 54.6% (lower layer) at high load compared to the inoculum, respectively.

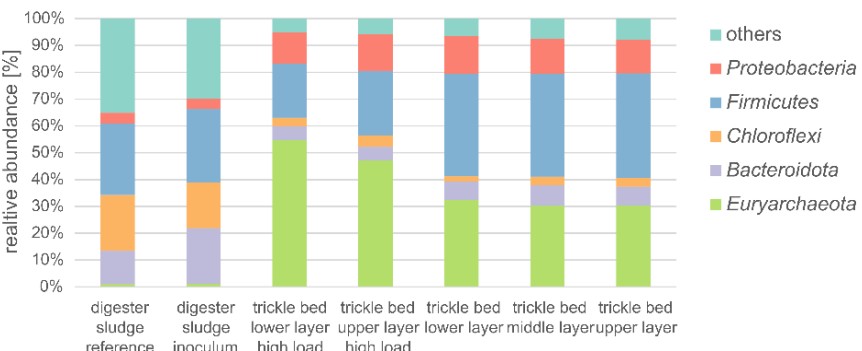

**Figure 9.** Overview of the microbial community dynamics analyzed by 16S rRNA gene amplicon sequencing displayed as relative abundance (%) on phylum level. Phyla with a relative abundance <10% are summarized as "others."

At both sampling time points, the relative abundance of *Euryarchaeota* was highest in the lower layer with 54.6% compared to 46.9% in the lower layer at high load (day 149) and 32.0% compared to 29.8% and 30.0% in the middle and the upper layer at (30.0%) of the trickle bed at lower $CO_2$ and $H_2$ feed rate (normal operation, day 102). As the phylum *Euryarchaeota* harbors the vast majority of known hydrogenotrophic methanogens [56], the observed pattern fits well with the available carbon dioxide and hydrogen as the substrates for hydrogenotrophic methanation.

In total, twelve different euryarchaeotal ASVs were detected in the samples, eleven ASVs belonging to the genus *Methanobacterium* and one to the genus *Methanobrevibacter.* In Figure 10, the four most abundant euryarchaeotal ASVs (>1% relative abundance) are displayed. *Methanobacterium* ASV_t63_7qa was barely present in the inoculum (relative abundance of 0.004%). In the biofilm of the trickling bed reactor, however, it was the most frequent ASV, with 23–28% from the upper layer to the lower layer at normal operation on day 102 and from 41–49% at high load operation (day 149) of the biofilm reactor. *Methanobacterium* ASV_t63_7qa showed 100% sequence identity with 100% query coverage to 43 different sequences deposited in NCBI's nt database. Among them were seven sequences classified as different strains of *Methanobacterium formicicum* obtained from enrichment cultures out of a biogas plant producing methane-rich gas in Germany described by Stantscheff, et al. [57]. Moreover, one of the most identical sequences to *Methanobacterium* ASV_t63_7qa belonged to the complete genome sequence of the hydrogenotrophic Archaeon *Methanobacterium* sp. Mb1 (accession number: HG425166.1) was isolated from a production-scale biogas plant, which was reported by Maus, et al. [58]. Besides, the relative abundance of two other euryarchaeal ASVs (*Methanobacterium* ASV_tqa_qpn, *Methanobrevibacter* ASV_d1n_8nj) also increased in the reactor but not nearly as dramatically (from 0.02–1.6% and 0.03–2.9%, respectively).

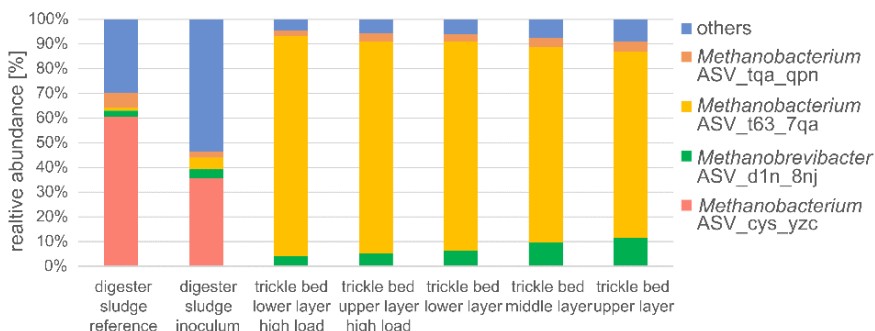

**Figure 10.** Dynamics of the euryarchaeotal ASVs obtained by 16S rRNA gene amplicon sequencing displayed as relative abundance (%) within the phylum Euryarchaeota. ASVs with a global relative abundance <1% are summarized as "others."

## 4. Conclusions

(1) An anaerobic trickling bed reactor was operated at mesophilic conditions for 225 days. The biological methanation of digester gas produced from municipal raw sludge was demonstrated under realistic conditions;

(2) Within 35 days after start-up, the methane concentration in the product gas reached 95% (65% in the raw gas);

(3) Two different biofilm carrier materials were tested, PU-foam cubes and structured packing media made of PE-hard plastic. Microscopic images showed a biofilm with a thickness of approximately 5–10 μm on both carrier types, whereby, depending on the gas load, more biomass growth occurs at the raw gas inlet in the lower part of the reactor;

(4) To promote the hydrogenotrophic pathway, $NH_4Cl$ was dosed to hold the $NH_4$-N concentration in the trickling liquid between 1500 mg/L and 3000 mg/L. At ammonium levels below 1000 mg/L, the acetic acid concentration increased, and it is assumed that the acetoclastic methanogens became dominant;

(5) The gas load was increased gradually, and stable methanation with methane concentration >95% was demonstrated at gas retention times between 2.5 h and 5 h;

(6) After adding additional biofilm carriers and two weeks of an operational pause without gas supply at mesophilic conditions, 21 days after starting the raw gas and hydrogen supply again, the methane concentration reached 95% again;

(7) Tests for trace elements and minerals (Na, K, Ca, Mg, Ni, Fe, and P) were used to determine biomass-specific uptake rates during the biological methanation process. Some trace elements such as Ni and Fe are required at low concentrations for the formation of hydrogenase enzymes independent of reactor load, some minerals are required load dependent (Na, K, Ca, and Mg), and for some elements, the concentration is strongly depending on the oxidation state (Fe, P). Remarkable is the load-dependent demand of the alkali (Na, K) and alkaline earth metals (Mg, Ca), which are needed in the metabolism of the archaea. For iron and phosphorus, a correlation between the oxidation state (FeII or FeIII) after opening the reactor for biomass sampling is assumed.

(8) For Na, K, Ca, Mg, and Ni biomass specific uptake rates were calculated, and a monthly dosage recommendation was given;

(9) Although microscopic examination of the biofilm did not reveal substantial differences between the different loads of the rector, the abundance and composition of the hydrogenotrophic biomass in the biofilm on the growing media changed dramatically. At high loadings, *Methanobacterium* sp. strongly dominated the microbial community with a relative abundance of up to 49%, while it was very rare in the inoculated sludge and the reference sludge with <0.05%.

**Author Contributions:** Conceptualization, J.T.; methodology, J.T., J.V. and D.M.; validation, J.T.; formal analysis, J.T.; investigation, J.T. and D.M.; resources, E.S., J.V. and J.T.; data curation, J.T.; writing—original draft preparation, J.T.; writing—review and editing, J.V., J.K. and K.S.; visualization, J.T.; supervision, K.S. and J.K.; project administration, J.T.; funding acquisition, J.K. All authors have read and agreed to the published version of the manuscript.

**Funding:** This research was funded by the Federal Ministry Republic of Austria Agriculture, Regions and Tourism, KPC Kommunalkredit Public Consulting, grant number B701315. The APC was funded by the TU Wien Bibliothek.

**Institutional Review Board Statement:** Not applicable.

**Informed Consent Statement:** Not applicable.

**Data Availability Statement:** The 16S rRNA gene amplicon sequencing data have been deposited at the Sequence Read Archive under the BioProject accession PRJNA947948.

**Acknowledgments:** This study is part of the project BioMAra—biological methanation in digesters in municipal wastewater treatment plants. The project was funded by KPC—Kommunalkredit Austria Public Consulting GmbH and the Federal Ministry of Austria for Sustainability and Tourism. We thank Zdravka Saracevic for her great effort in the lab, "Abwasserverband Grossraum Bruck an der Leitha—Neusiedl am See" for providing raw and digester sludge for the lab-scale tests, and we thank the JMF, particularly Julia Ramesmayer, Petra Pjevac and Joana Séneca for sample preparation for microbiome analysis, 16S rRNA gene amplification, sequencing, and their constructive feedback.

**Conflicts of Interest:** The authors declare no conflict of interest. The funders had no role in the design of the study; in the collection, analysis, or interpretation of data; in the writing of the manuscript, or in the decision to publish the results.

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
