# Peer review of "Biological Methanation in an Anaerobic Biofilm Reactor—Trace Element and Mineral Requirements for Stable Operation"

_processes, doi:10.3390/pr11041013_

Round 1

Reviewer 1 Report

Review on the paper "Biological Methanation in an Anaerobic Biofilm Reactor - Trace Element Requirements for stable Operation" by Joseph Tauber, Daniel Möstl, Julia Vierheilig, Ernis Saracevic, Karl Svardal and Jörg Krampe.

This paper is very topical and potentially useful for many applications. As for me, the main result is formulated as follows (Conclusions 7 and 8)

"Using tests for trace elements (Na, K, Ca, Mg, Ni, Fe, and P) biomass specific uptake rates were found. Some trace elements like Ni are needed independently from the reactor load, some are needed load depending (Na, K, Ca, and Mg), and for some elements, no correlation between load and consumption was found (Fe, P). For iron and 

phosphorus, a correlation between the oxidation state (FeII or FeIII) after opening the reactor for biomass sampling is assumed.for Na, K, Ca, Mg, and Ni biomass specific uptake rates were calculated and a monthly dosage s. was given".

As mentioned, this results is very useful for industrial and environmental application.

Therefore, I recommend this paper for publication in "Processes"

However, the same conclusions reflect the obvious drawback, i.e., in the paper there is no attempt of explaining this fact, at least the qualitative one. So, the result is just empirical. 

No doubt, the biological methantion is the extremely complex process. However, in conclusions, there is a hidden indication of the physico-chemical nature of the phenomenon. The group of Na, K, and Ca and Mg ("needed load depending") belongs to alkali metals (Group I) and alkaline earth (Group II) metals, respectively. For these metals, the oxidation state is well defined. I recommend authors to formulate the essential qualitative explanation of this effect  and mention it within the body of the paper, and in conclusions as well.

My recommendation is "minor revision".

Author Response

Thank you for your review, which helped improve the paper, especially considering Group I and II elements. I agree in all aspects. Please find the point-by-point answers in the attached word file.

Thank you and all the best,

Joseph Tauber

Reviewer 2 Report

Good article, but it will be interesting if the introduction could be more objective.

Author Response

Answer to Review 2: Thank you for your review, some changes were made in the introduction, and some references were added. Please see the changes in the text.

Thank you and all the Best,

Joseph Tauber

Reviewer 3 Report

Authors investigated process of biological methanation via trickling bed reactor. Influence of composition of trickling liquid on operational condition was studied. Paper is quite large and complex, but some clarification and correction could improve the quality of study.

Lines 35-36. The authors point out “…biomethane, which can be used as a substitute for natural gas…”. Are there any objective technical barriers to doing this substitution? Additional clarifications are needed.

Line 38. “Bevor” – is a typo.

Line 47. At the discretion of the authors, the abbreviation “Mio” can be replaced, as this is a rare form of abbreviation and may confuse the reader.

Line 48. The acronym “WWTP” requires an explanation at the first mention.

Lines 92-93, 130, 168. “Media/milieu” is jargon, the profile of this journal is more general, perhaps neutral “medium/environment” should be used.

Line 199. What is “The trickling liquid”? It is described in detail in the manuscript, but what it means standalone is not clear.

Line 204. When describing the installation, it is probably not worth giving such wide ranges of parameters “…850-2800 m2/m3…”.

Lines 236-238. Taking into account a larger number of set-up elements, including a hydrogen generator, a small explanation should be given regarding the economic component of the approach, or at least the power consumption per unit volume of the finished product should be given. The only clue to hydrogen dosing is the rotameter, with a measuring range of 0-35 ml/min. Thus, a considerable amount of the most energetically valuable hydrogen is used for the production of

Line 326. Table 5. Standard deviation is 2 times greater than the value. Clarification needed here.

Fig. 2. The figure is overloaded, concentrations that go beyond 100% cause questions about the error of the measurement instrument.

Figure 4. Is it appropriate, according to the authors, to present linear dependences for changing concentrations, in the case when there are too many influencing factors and parameters during the operation of a complex installation?

Fig 7,8. Scale needs to be provided.

Author Response

Review 3

--> Answer to review 3: Thank you for your very accurate review! I agree in all aspects and hereby respond point by point. Please also see the changes in the text attached.

Thank you and all the best,

Joseph Tauber

Comments and Suggestions for Authors

 Authors investigated process of biological methanation via trickling bed reactor. Influence of composition of trickling liquid on operational condition was studied. Paper is quite large and complex, but some clarification and correction could improve the quality of study.

Lines 35-36. The authors point out “…biomethane, which can be used as a substitute for natural gas…”. Are there any objective technical barriers to doing this substitution? Additional clarifications are needed.

-->Agree, added a sentence: ..after drying, desulfurization and removal of impurities such as dust and siloxane. Due to different legal frameworks, the quality requirements for grid injection vary between different countries (Munoz et al. 2015).

Line 38. “Bevor” – is a typo.

-->Agree Changed spelling from Bevore to Before, thank you

Line 47. At the discretion of the authors, the abbreviation “Mio” can be replaced, as this is a rare form of abbreviation and may confuse the reader.

-->Reviewer 3 --> Agree, changed Mio to ·106

Line 48. The acronym “WWTP” requires an explanation at the first mention.

-->Agree, added: wastewater treatment plants (WWTPs)

Lines 92-93, 130, 168. “Media/milieu” is jargon, the profile of this journal is more general, perhaps neutral “medium/environment” should be used.

-->Agree --> changed sentence, milieu exchanged for environment

Line 199. What is “The trickling liquid”? It is described in detail in the manuscript, but what it means standalone is not clear.

-->Agreeà added sentence to make it clear (line 204-207):

Line 204. When describing the installation, it is probably not worth giving such wide ranges of parameters “…850-2800 m2/m3…”.

-->Agree--> changed sentence, removed reference with large range values.

Lines 236-238. Taking into account a larger number of set-up elements, including a hydrogen generator, a small explanation should be given regarding the economic component of the approach, or at least the power consumption per unit volume of the finished product should be given. The only clue to hydrogen dosing is the rotameter, with a measuring range of 0-35 ml/min. Thus, a considerable amount of the most energetically valuable hydrogen is used for the production of

-->Agree--> A sentence was added :

The hydrogen was introduced in a stoichiometric ratio of 4:1 to the carbon dioxide as shown in equation 2. The energy requirement for the production of methane via biological methanation, including efficiencies can be given as 22.5 kWh/m3 methane, according to calculations by Tauber et al. (2021), not taking into account the oxygen produced via electrolysis An efficiency of 49% - 79% for the whole power to gas process via biological methanation is indicated by Sterner and Stadler (2014) depending on the pressure level, storage type and technology used.

Line 326. Table 5. Standard deviation is 2 times greater than the value. Clarification needed here.

-->Agree--> To be more clear, changed form mean+/- SD to mean and min/max values for H2S because of the large range of values.

Fig. 2. The figure is overloaded, concentrations that go beyond 100% cause questions about the error of the measurement instrument.

-->Agreeà the measurement error is now provided in the material and methods section. A sentence was added: , the measurement accuracy is specified as 3% by the manufacturer at 100% CO2 and CH4.

Figure 4. Is it appropriate, according to the authors, to present linear dependences for changing concentrations, in the case when there are too many influencing factors and parameters during the operation of a complex installation?

-->Agree--> As far as possible for a complex biological system, all influencing factors were kept constant (temperature, load, etc.). R2 was removed from figure 4 (not necessary), but the calculated slope for the different experimental phases was left.

Fig 7,8. Scale needs to be provided.

-->The scale is now displayed separately in each microscopic image (200 µm/500 µm). All Microscopic images including a new clearer scale, were newly inserted into Figure 7 and Figure 8.

Submission Date

21 February 2023

Date of this review

09 Mar 2023 09:40:50

Reviewer 4 Report

1. Discussion should be improved and appropriate citations should be included in the Organic Acid Concentrations in the Trickling Liquid section. 2. Like Euryarchaeotal ASVs (Figure 10), Proteobacteria, Firmicutes, Chloroflexi, bacteriodota should be explained. 3. For the reader's easy understanding, I suggest the authors include the trace elements for dosing in the trickling liquid of the present study in Table 2. 4. Provide the sequence of the primers used for amplicon sequencing.

Author Response

--> Answer to review 4:

Thank you for your accurate review! I agree in all aspects and hereby respond point by point. Please see the changes in the manuscript attached.

Thank you and all the best,

Joseph Tauber

Comments and Suggestions for Authors

  1. Discussion should be improved and appropriate citations should be included in the Organic Acid Concentrations in the Trickling Liquid section.

-->Agree-->A Reference was added and a sentence was added.

  1. Like Euryarchaeotal ASVs (Figure 10), Proteobacteria, Firmicutes, Chloroflexi, bacteriodota should be explained.

-->Answer: Because Euryarcharchaeota contains the species which are interesting for the hydrogenotrophic conversion, they are presented separately. Proteobacteria, Firmicutes, Chloroflexi, and Bacteriodota are not contained in the Manuscript, but the microbiome sequencing data are available under the BioProject accession number PRJNA947948

  1. For the reader's easy understanding, I suggest the authors include the trace elements for dosing in the trickling liquid of the present study in Table 2.

-->Partly agree--> I was thinking also to present recipes from the literature together with the one used in this study, which would be easier to understand/compare. But it is not possible because of the space (table width) and because of the mixing of introduction/literature and material and methods. That is why we have decided to keep them strictly separated.

  1. Provide the sequence of the primers used for amplicon sequencing.

Agree--> primer sequences were added for 515F and 806R primers

primers 515F (5’-GTG CCA GCM GCC GCG GTA A-3’) (Parada et al. 2016) and 806R (5’-GGA CTA CNV GGG TWT CTA AT-3’) (Apprill et al. 2015)

Submission Date

21 February 2023

Date of this review

18 Mar 2023 13:07:27
